# LADDERSYM: A MULTIMODAL INTERLEAVED TRANSFORMER FOR MUSIC PRACTICE ERROR DETECTION

**Benjamin Shiue-Hal Chou**[1]**, Purvish Jajal**[1]**, Nick John Eliopoulos**[1]**, James C. Davis**[1]**,
George K. Thiruvathukal**[2]**, Kristen Yeon-Ji Yun**[1]**, Yung-Hsiang Lu**[1]
[1]Purdue University   [2]Loyola University Chicago

{chou150, pjajal, neliopou, davisjam, yun98, yunglu}@purdue.edu
gkt@cs.luc.edu

## ABSTRACT

Music learners can greatly benefit from tools that accurately detect errors in their practice. Existing approaches typically compare audio recordings to music scores using heuristics or learnable models. This paper introduces *LadderSym*, a novel Transformer-based method for music error detection. *LadderSym* is guided by two key observations about the state-of-the-art approaches: (1) late fusion limits inter-stream alignment and cross-modality comparison capability; and (2) reliance on score audio introduces ambiguity in the frequency spectrum, degrading performance in music with concurrent notes. To address these limitations, *LadderSym* introduces (1) a two-stream encoder with inter-stream alignment modules to improve audio comparison capabilities and error detection F1 scores, and (2) a multimodal strategy that leverages both audio and symbolic scores by incorporating symbolic representations as decoder prompts, reducing ambiguity and improving F1 scores. We evaluate our method on the *MAESTRO-E* and *CocoChorales-E* datasets by measuring the F1 score for each note category. Compared to the previous state of the art, *LadderSym* more than doubles F1 for missed notes on *MAESTRO-E* (26.8% → 56.3%) and improves extra note detection by 14.4 points (72.0% → 86.4%). Similar gains are observed on *CocoChorales-E*. Furthermore, we also evaluate our models on real data we curated. This work introduces insights about comparison models that could inform sequence evaluation tasks for reinforcement learning, human skill assessment, and model evaluation. [1]

## 1 INTRODUCTION

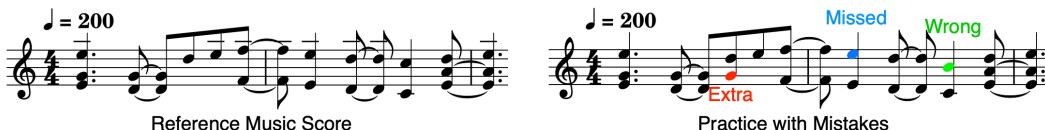

Figure 1: *The error detection task for music practice.* Given a reference score and a performance, solutions must detect three types of errors: *extra notes*; *missed notes*; and *wrong notes*. Wrong notes are represented as both a missed note and an extra note.

Effective skill assessment accelerates human skill acquisition by providing targeted feedback (Sigrist et al., 2013). Music learners, in particular, benefit from tools that surface practice mistakes (Apaydınlı, 2019). Yet over 4 million U.S. K–12 students lack access to music education (Morrison et al., 2022), highlighting the need for accessible feedback tools. Music practice error detection addresses this need by comparing a practice recording to a reference music score, as illustrated in Figure 1, where the score may be provided in symbolic or audio form.

Commercial apps for music education, such as Yousician (YousicianLtd., 2024) and Simply Piano (JoyTunes, 2024), are widely used with over 10 million downloads each. However, these systems

---

[1]Code: https://github.com/ben2002chou/LadderSYM

offer only coarse correctness judgments (*e.g.*, whether a note is correct) and do not classify error types such as missed, extra, or wrong notes. This limits the quality of feedback. In contrast, recent research attempts to provide finer-grained feedback by aligning student practice audio with symbolic reference scores (Benetos et al., 2012; Wang et al., 2017). Chou et al. (2025) found that such alignment-based methods break down when performance deviates substantially from the reference, limiting their reliability for error detection. To address this, Chou et al. (2025) adapted transformers (Vaswani et al., 2017), achieving superior F1 scores. Their model compares practice and reference recordings in the latent space, eliminating the need for explicit alignment. Despite these advances, we observe two key limitations in this state-of-the-art approach. (1) The model uses late fusion by combining the two audio streams with a single joint encoder layer. Through ablations and attention map visualizations, we show that this design limits alignment capacity. Stacking multiple joint layers improves alignment but restricts asymmetric feature extraction. (2) The score is represented only as audio. This introduces ambiguity about which notes are present, especially when multiple notes are played at once. Overlapping frequency content makes it difficult to distinguish individual notes.

We introduce *LadderSym*, a new architecture that addresses both limitations. To address limitation (1), we design **Ladder**,[2] a two-stream encoder that shifts alignment to inter-stream alignment modules via inductive bias. This allows standard transformer encoder layers to focus on feature extraction without being forced to share capacity for alignment. To address limitation (2), we incorporate both audio and symbolic representations of the score. The symbolic score, denoted as **Sym**, refers to the tokenized version of the music score. This symbolic score is provided to the decoder as a prompt, while the audio score is processed through the encoder and serves as context. This design reduces ambiguity of score inputs.

*LadderSym* achieves **state-of-the-art** performance on both the synthetic *MAESTRO-E* and *CocoChorales-E* datasets. On *MAESTRO-E*, it more than doubles F1 for Missed Notes ($26.8\% \rightarrow 56.3\%$) and improves Extra Note detection by **14.4 points** ($72.0\% \rightarrow 86.4\%$). On *CocoChorales-E*, it improves Missed Note F1 from **51.3%** to **61.7%**, and Extra Note F1 from **46.8%** to **61.4%**. We also validate our model's generalization on a new, largest to date dataset we curated of real-world beginner performances. Demo examples of model outputs are available at: our demo page.

Our contributions are:

1. We develop a novel encoder architecture that improves comparison by aligning audio representations frequently across input streams (§3.1).
2. We improve model performance with a multimodal strategy, by prompting the decoder with symbolic music inputs and reducing the ambiguity of its inputs (§3.2).
3. We analyze transformer attention patterns to extract design principles for cross-modal comparison and apply them to improve model performance (§4.3.1, Appendix A.9).

***Significance:*** The high accuracy of *LadderSym* takes a step toward solving the field's "chicken-and-egg" data problem (Kirillov et al., 2023; Monarch, 2021): its potential as an assistive annotation tool can help create real-world datasets needed to train the next generation of models(see §A.7.2 for our human-in-the-loop workflow). Annotating just 20 pieces required approximately 52 person hours, and scaling further would demand hundreds more. More broadly, our work focuses on music practice error detection, which is fundamentally an evaluation task that involves aligning and comparing two inputs. The architectural insights from this paper could inform the design of more effective evaluation methods in other domains, such as reinforcement learning, other human-skill assessment, and model evaluation. We elaborate on these implications in §5.

## 2 Background and Related Work

We review music practice error detection (§2.1) and common multimodal design strategies (§2.2).

### 2.1 Error Detection for Music Practice

Music error detection (Figure 1) is an instance of the sequence-to-sequence learning problem (Sutskever et al., 2014; Luong et al., 2016; Hawthorne et al., 2021). Specifically, it is a

---

[2]The name reflects our goal to help students "climb the ladder" of music skill development.

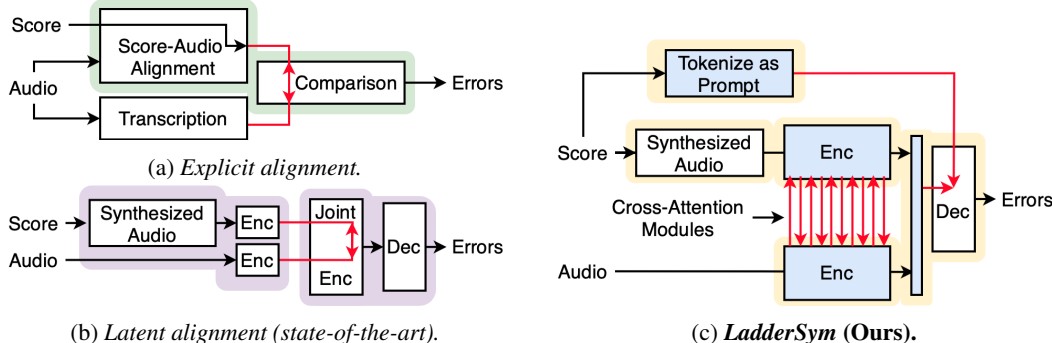

Figure 2: *Comparison of error detection architectures.* (a) Explicit alignment methods align the score with audio and compare it to the transcribed practice (Benetos et al., 2012). (b) Latent alignment methods synthesize the score to audio and pass it to the encoder directly, without explicit alignment (Chou et al., 2025). (c) Our method, *LadderSym*, is a latent alignment approach that incorporates symbolic score prompting to address score ambiguity and enhanced cross-stream information flow via cross attention modules.

many-to-one sequence translation task, as it relies on two sequences (practice and reference, audio or symbolic) that are compared to produce an error sequence. There are two main approaches to music practice error detection: explicit alignment and latent alignment, illustrated in Figure 2.

*Explicit alignment* methods compare transcribed notes from the score and practice audio (Figure 2a). Techniques such as Dynamic Time Warping (DTW) (Sakoe & Chiba, 1978) align the score and practice audio to facilitate this comparison. These methods identify differences by comparing the symbolic score to reference audio (Benetos et al., 2012; Wang et al., 2017). However, DTW is sensitive to deviations from the reference sequence, commonly present in practice recordings with errors, leading to inadequate error detection (Chou et al., 2025).

In contrast, *latent alignment* methods forgo explicit sequence alignment and instead learn to directly output mistakes by comparing the score and practice streams (Figure 2b). *Polytune* (Chou et al., 2025) pioneered this direction, coupling an Audio Spectrogram Transformer encoder with a T5 decoder to align synthesized score audio with the practice recording. Latent alignment delivered clear gains: compared to an explicit-alignment baseline, *Polytune* raised Missed-note F1 on *MAESTRO-E* from 6.6% to 26.8% and Extra-note F1 from 39.9% to 72.0%. Although *Polytune* is the state of the art, its performance is still low on missing notes, and its alignment behavior is not yet well understood.

Our proposed method, ***LadderSym*** (Figure 2c), builds upon the latent alignment paradigm but introduces two key innovations to address these limitations. First, as shown in the top-right of panel (c), we incorporate a symbolic representation of the score as a prompt to the decoder. This provides a clear, non-ambiguous reference that bypasses the overlapping frequency issues inherent in audio-only scores. Second, we introduce a novel encoder with inter-stream alignment modules. Unlike the late-fusion approach in *Polytune*, which aligns streams only in the final layer, our architecture allows for the decoupling of feature extraction from alignment via an alignment module at every layer.

## 2.2 DESIGN OF MULTIMODAL ENCODERS

Our work addresses these limitations by drawing on principles from multimodal encoder design. Multimodal models handle multiple input modalities or different representations of the same modality (*e.g.*, RGB and depth maps). They use either a single-stream encoder (early fusion) or separate, parallel encoders with fusion layers that enable cross-modal interaction. Baltrusaitis et al. (2019) identify three dominant paradigms for such fusion: early, late, and hybrid. Early and late fusion appear more often when training from scratch (early fusion (Girdhar et al., 2022; Li et al., 2019); late fusion (Ao et al., 2022; Gong et al., 2023; Akbari et al., 2021; Girdhar et al., 2023; Radford et al., 2021; Chen et al., 2023), hybrid fusion (Alayrac et al., 2022; Li et al., 2022)).

Hybrid fusion occupies the middle ground between early and late fusion. One approach is to use cross-attention to condition an encoder on the output of another encoder. Recently, hybrid designs

most commonly arise when adapting language models (LMs) for vision tasks. The prevailing strategy is to pair a pre-trained Vision Transformer (ViT) with a (usually frozen) language model and condition each LM layer on external modality information (Alayrac et al., 2022; Li et al., 2022), with an adapter inserted between the final vision encoder layer and every LM layer.

In this paper, we introduce a novel hybrid fusion approach that encourages alignment to be handled by inter-stream alignment modules while decoupling feature extraction across modalities. Our design still uses cross-attention since it can be used to pass information between token streams of varying length with minimal impact on latency (Jajal et al., 2024; Koner et al., 2024), but it diverges from two common hybrid strategies. First, unlike conditioning approaches such as CLIP (Radford et al., 2021), where interaction often occurs just once on a compressed, static vector, our model performs *iterative, layer-wise cross-attention on the full sequence of features*. Second, this allows our architecture to *implicitly learn a fine-grained temporal alignment*. This contrasts with hybrid temporal models that assume alignment *a priori*, for instance by interleaving multimodal tokens at the input layer (Xu et al., 2025). The resulting learned alignment mechanism is key for complex comparison tasks like music error detection and may generalize to other fine-grained comparison problems, as we discuss in §5.

## 3 METHOD

This section details *LadderSym*, our multimodal transformer for music error detection. Our design revolves around two questions raised by the analysis in §2.1: (1) where should alignment occur between score and practice streams? (2) how can we reduce the ambiguity introduced by representing the score solely as audio?

To answer these questions we study the behavior of existing architectures. We probe the representations learned by late- and early-fusion encoders, quantify how fusion depth affects performance, and analyze how decoder inputs influence missed-note detection (Tables 1 and 4). These measurements clarify how much cross-stream interaction is necessary and highlight the information lost when the decoder lacks symbolic context.

Guided by these findings, we construct *LadderSym*, shown in Figure 3. The **Ladder** encoder is a two-stream transformer whose cross-attention modules precede each layer, aligning score and practice audio representations while decoupling feature extraction from alignment (§3.1). The **Sym** prompt supplies symbolic score tokens to the decoder (§3.2), reducing reliance on ambiguous audio-only cues and improving detection of subtle errors such as missed notes. We summarize model I/O, architectural configurations, and other implementation details in §3.3.

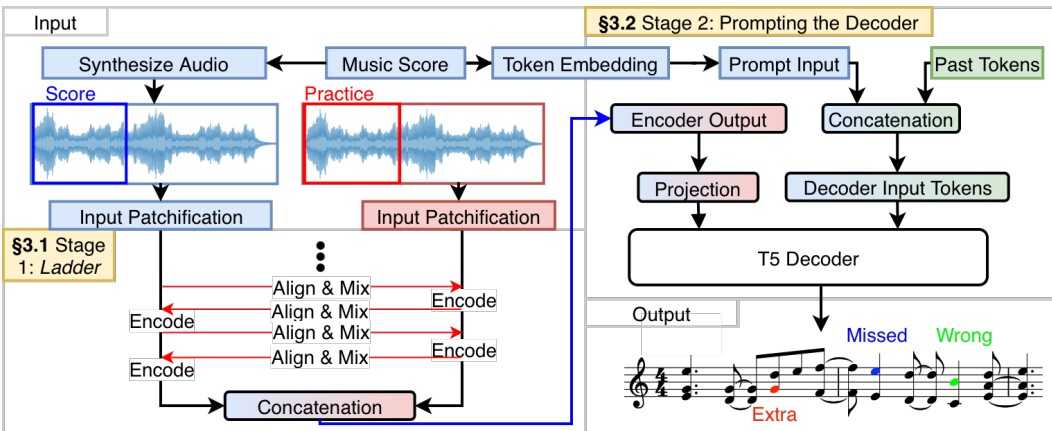

Figure 3: *Architecture of LadderSym.* We feed both *score audio* and *practice audio* into *Ladder*, our *novel encoder* with inter-stream alignment modules. Their latent features are concatenated and used as context for the autoregressive decoder. To create *LadderSym*, we prepend a symbolic prompt that is generated from a MIDI version of the score before the start-of-sequence token to provide a different representation of the reference score. The T5 decoder then produces MIDI-like tokens, labeling notes as correct, missed, or extra.

### 3.1 Stage 1: The Ladder Encoder

***Motivation:*** *Ladder* aims to overcome a key limitation of the state-of-the-art *Polytune* architecture: its late fusion design lacks interaction between the practice and score inputs. Our ablations show that fusing earlier enhances interaction between reference and practice inputs, improving performance (Table 4). Thus, we conclude that inter-stream information flow is beneficial to our model's ability to compare the inputs. However, fusing too early harms performance, as shown when using more than three joint encoders (Table 4). Attention maps (Appendix A.9) reveal that late fusion limits alignment and comparison between inputs. Early fusion enables alignment in initial layers but sacrifices cross-stream feature extraction due to parameter sharing (Table 1). To guide our encoder design, we first probe (Raghu et al., 2021) the latent representations of two baseline architectures: *Polytune* and an early fusion encoder. Each encoder is frozen, and we train probes to evaluate locality and globality of the learned representations. Locality is measured by whether each stream retains token-level temporal position information and globality is measured by coarse clip-level energy information. In *Polytune*, the practice stream maintains strong locality (0.86), while the score stream shows reduced local accuracy (0.45) but improved global awareness ($0.179 \rightarrow 0.186$). This pattern suggests a division of labor: one stream specializes in local detail, while the other encodes global features. In contrast, the early fusion encoder yields high locality in both streams (0.91 and 0.93), along with balanced global information. This is because both streams share parameters. We hypothesize that this limits the streams' ability specialize, which intuitively can harm comparison performance, as comparing $A$ to $B$ should be redundant with comparing $B$ to $A$.

This motivates a design that combines the strengths of early and late fusion. Our model is similar to late fusion in that its design uses separate encoders for each input stream. One encoder extracts local features, while the other captures global features. Unlike late fusion, our architecture supports interaction at each layer for fine-grained alignment, similar to early fusion. This novel design enables effective cross-modal comparison without being constrained by cross-modal parameter sharing.

***Architecture:*** Our design for *LadderSym* consists of a novel interleaved encoder architecture, which we designed based on intuitions about comparison tasks. Before each transformer block, one input stream is aligned and additively fused into the other. The cross-attention alignment module (1) enables alignment at each layer and (2) allows the transformer blocks to focus on feature extraction. As shown in Figure 4, the learned attention maps recover the same off-diagonal structure as DTW alignments, demonstrating that corresponding tokens across time learn to attend to each other. Attention maps for deeper layers are shown in Figure 8.

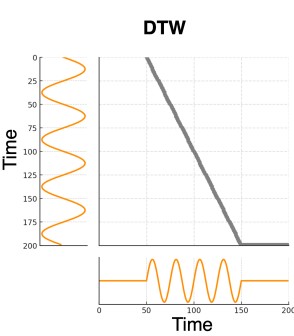
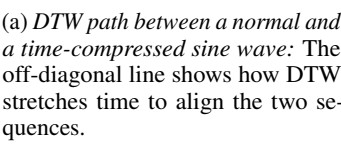
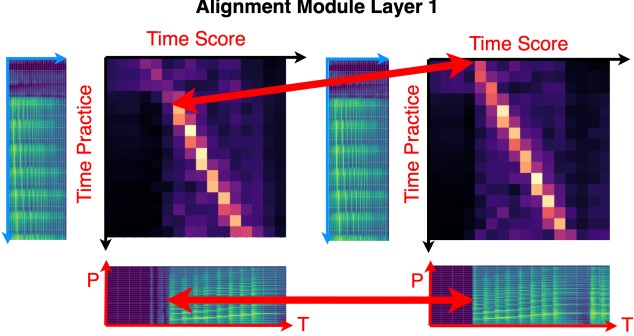

(a) *DTW path between a normal and a time-compressed sine wave:* The off-diagonal line shows how DTW stretches time to align the two sequences.

(b) *Cross-attention map from the alignment module:* The x- and y-axes denote time in the score and practice spectrograms, respectively. Attention map values are averaged over the pitch dimension (P) to highlight temporal alignment (T). For the attention map on the right, we shift the score forward by 0.5 seconds. We can see that the model's attention shifts to the upper left.

Figure 4: *Similarity between (a) Dynamic Time Warping and (b) learned alignment patterns in the alignment module.* This visual similarity suggests that the alignment module is successfully learning a meaningful temporal correspondence between the two audio streams, analogous to the explicit alignment path found by classical algorithms like DTW.

The process for one encoder block is described by:

$$\mathbf{P}_{\text{ref}}^{(i+1)} = \text{ViT}_{\text{ref}}\Big(\mathbf{P}_{\text{ref}}^{(i)} + \mathbf{CA}\big(\mathbf{P}_{\text{prac}}^{(i)}, \mathbf{P}_{\text{ref}}^{(i)}\big)\Big), \tag{1}$$

$$\mathbf{P}_{\text{prac}}^{(i+1)} = \text{ViT}_{\text{prac}}\Big(\mathbf{P}_{\text{prac}}^{(i)} + \mathbf{CA}\big(\mathbf{P}_{\text{ref}}^{(i+1)}, \mathbf{P}_{\text{prac}}^{(i)}\big)\Big). \tag{2}$$

Here, $\mathbf{P}_{\text{ref}}$ represents the score audio embeddings, $\mathbf{P}_{\text{prac}}$ the practice audio embeddings, $\mathbf{CA}$ the cross-attention operation, and $i$ the current layer index. The final fused representation is obtained as:

$$\mathbf{H}_{\text{fused}} = \text{Concat}(\mathbf{P}_{\text{ref}}^{(\text{final})}, \mathbf{P}_{\text{prac}}^{(\text{final})}). \tag{3}$$

The alignment module combines cross-attention and additive fusion to to first align then pass information between streams at each layer. Additive fusion means directly adding cross-attention output to the stream embedding, preserving both self and aligned information from the other stream. This fused representation is then processed by a standard ViT block. We then reverse the alignment and fusion direction and pass the result through the next ViT block. Finally, we concatenate both final states into a fused latent $\mathbf{H}_{\text{fused}}$.

We illustrate the encoder block in Figure 5. The alignment module is summarized by the expression:

$$\mathbf{P}_{\text{prac}}^{(i)} + \mathbf{CA}(\mathbf{P}_{\text{ref}}^{(i+1)}, \mathbf{P}_{\text{prac}}^{(i)}),$$

where the cross-attention output is directly added to the current representation before being passed to the next encoder block.

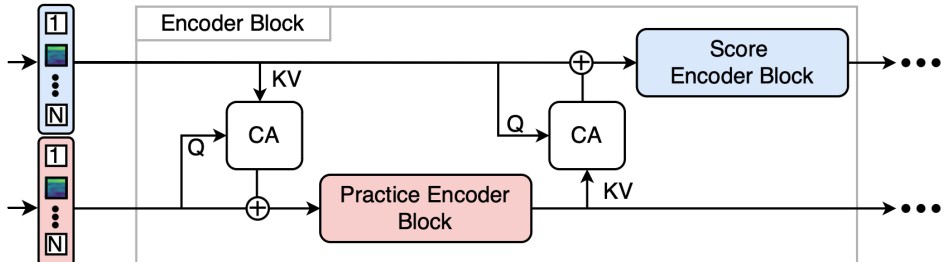

Figure 5: *Topology of the encoder block:* Alignment modules alternate between streams, allowing iterative alignment and fusion of information from the score and practice audio. The encoder blocks process the intermediate representations.

Table 1: *Probing frozen encoders.* To analyze the representations learned by each encoder, the model is frozen and evaluated using lightweight linear probes. Locality predicts a token's temporal position, globality predicts a clip-level energy label, and cross-stream correspondence uses one stream to predict the other's energy. For *Polytune*, we extract features before the joint encoder layer, so cross-stream probes are not applicable. All values are accuracies (see Appendix A.1 for setup details).

| **Model** | Local score | Local practice | Global score | Global practice | Cross-Stream practice to score | Cross-Stream score to practice |
|-----------|-------------|----------------|--------------|-----------------|--------------------------------|--------------------------------|
| *Polytune* | 0.452 | 0.862 | 0.186 | 0.179 | – | – |
| Early fusion | **0.909** | **0.931** | **0.292** | **0.273** | **0.280** | 0.269 |
| *LadderSym* | 0.197 | 0.681 | 0.162 | 0.252 | 0.158 | **0.300** |

**Probing Ladder:** Having introduced the interleaved encoder, we return to the probing framework to assess how this design shapes latent representations. Using the same probing setup, we now evaluate *LadderSym* in terms of locality, globality, and cross-stream correspondence. Results are shown in Table 1. We find that the practice stream retains strong local information (0.681), and the score stream has reduced locality (0.197). However, the score stream encodes cross-stream features from the practice stream more accurately than any prior model (0.30). These results confirm that *LadderSym* supports an asymmetric division of labor between streams, enabling specialization.

## 3.2 Stage 2: Harnessing Symbolic Scores by Prompting the Decoder

We give the decoder direct access to symbolic score information via prompting to leverage the complementary strengths of symbolic and audio representations. Symbolic-only tokenizers can introduce alignment errors, especially in complex time signatures (Fradet et al., 2021), while audio representations often suffer from overlapping frequency bins that obscure concurrent notes. Providing both views helps mitigate these respective weaknesses. Table 5 shows that using our prompting strategy on *Polytune* (Prompt + Audio) can significantly improve performance over just using audio inputs (Audio Only). We also test a variation of *Polytune* and find that Audio Only outperforms using only the prompt (Prompt Only). Table 5 also shows that combining our prompted decoder strategy with the encoder that includes inter-stream alignment modules yields the highest scores for all categories in MAESTRO-E and for missed notes in CocoChorales-E. We use a T5 decoder following Chou et al. (2025).

## 3.3 Implementation Details

***Input/Output Format:*** To tokenize the input audio spectrogram, we follow the procedure in MT3 (Gardner et al., 2022) and *Polytune*. The output format also follows (Chou et al., 2025), which is a modified version of (Gardner et al., 2022), omitting instrument tokens (assuming a single-instrument setting) and adding explicit error labels (`extra`, `missed`, `correct`). Further details for both are presented in Appendix A.2 and Appendix A.3. A detailed breakdown of our prompt tokenization scheme is provided in Appendix A.2.2 for interested readers.

***Model Implementation:*** *LadderSym* has a configuration of 12 transformer encoder layers and 8 decoder layers to match the layer count of the AST (Audio Spectrogram Transformer) (Gong et al., 2021) encoder and T5 decoders used in (Chou et al., 2025). The transformer encoder output, with a dimensionality of 768, is projected down to 512 to match the T5 decoder's dimensionality. Our training regime follows that of (Chou et al., 2025) and is detailed in Appendix A.4. We use the same hyperparameters as *Polytune*. We adapted model component code from `EfficientTTMs` (Jajal et al., 2024)(MIT License), though our approach differs in design. We also build on `Polytune` (BSD 3-Clause, non-commercial).

## 4 Results

We present results comparing *LadderSym* against *Polytune* (Chou et al., 2025) and an explicit-alignment baseline derived from Benetos et al. (2012) and Wang et al. (2017). Evaluations cover *CocoChorales-E*, *MAESTRO-E* (Table 2), and a small real-world dataset. Full experimental details are in §4.1. Our evaluation includes: (1) A quantitative comparison showing improved F1 scores on all datasets (§4.2); (2) an ablation study analyzing different variants of the encoder (§4.3.1); and (3) an ablation study examining the effect of prompting with the symbolic music score (§4.3.2).

## 4.1 Experimental Design

***Software and Hardware:*** We train and evaluate our models using PyTorch 2.3.0 and Transformers 4.40.1 on an NVIDIA A100-80GB GPU.

***Datasets:*** We follow Chou et al. (2025) and evaluate on their partially synthetic benchmarks, *MAESTRO-E* (competition piano repertoire with dense chords) and *CocoChorales-E* (13 single-instrument tracks without overlap). Dataset construction and summary statistics appear in Appendix A.6. All ablations use the combined synthetic test split (4401 tracks).

To test real-world robustness, we curated a 20-track beginner piano set: Three graduate students who are novice pianists each played 6-7 pieces. To our knowledge, this is the first dataset of genuine beginner mistakes. Each take pairs the reference MIDI score with practice audio and note-level annotations verified by two musically trained students. Per-piece metrics and additional details are in Appendix A.7. Every model is additionally evaluated on this real-data set without finetuning.

***Explicit-alignment baseline:*** This baseline extends Benetos et al. (2012) and Wang et al. (2017) with the more modern MT3 Gardner et al. (2022). Implementation details appear in Appendix A.8.

Table 2: *Comparison of LadderSym and Polytune across error types in two datasets, MAESTRO-E and CocoChorales-E.* Error types are abbreviated (C = Correct, M = Missed, E = Extra); CocoChorales metrics are macro-averaged across instruments. Blue highlights mark *LadderSym*, pink highlights mark *Polytune*, and bold indicates the best value per column; colors are for visual emphasis only. The explicit-alignment baseline appears in the rightmost block for context.

| | *LadderSym* | | | *Polytune* | | | Explicit Align. | | |
|---|---|---|---|---|---|---|---|---|---|
| **Dataset** | C | M | E | C | M | E | C | M | E |
| *MAESTRO-E* | **94.4%** | **54.7%** | **86.4%** | 90.1% | 26.8% | 72.0% | 43.5% | 6.6% | 39.9% |
| *CocoChorales-E* | **97.7%** | **61.7%** | **61.4%** | 95.4% | 51.3% | 46.8% | 36.7% | 7.7% | 23.5% |

*Metrics:* We use the evaluation metric Error Detection F1, introduced by (Chou et al., 2025). Error Detection F1 measures the F1 score for *Missed*, *Extra*, and *Correct* notes.

## 4.2 QUANTITATIVE RESULTS

*Main results:* F1 scores for CocoChorales-E and MAESTRO-E are presented in Table 2, with per-instrument results available in Appendix A.10. We achieve across-the-board improvements. As expected, on the highly concurrent MAESTRO-E dataset, the performance gain is most notable, with the missed note F1 score improving from 26.3% to 54.7%.

*Real-World Performance:* To validate our model's effectiveness beyond synthetic data, we also tested it on a new, manually annotated dataset of real-world beginner performances. The model was not finetuned. On this authentic data, *LadderSym* shows a marked improvement in detecting missed notes (78.5% vs. 63.9% F1) and a small gain in extra note detection (81.6% vs. 80.6% F1) when compared to *Polytune*. While the overall performance gap is smaller due to the pieces being simpler than MAESTRO-E, *LadderSym* still has better across the board performance. Most importantly, the results demonstrate that *LadderSym*'s architectural improvements successfully generalize to genuine human errors. More details are in the Appendix (Appendix A.7).

*Model size and speed:* *LadderSym* (172M parameters) is more efficient than the state-of-the-art *Polytune* (192M parameters) despite its iterative alignment modules. By comparison, under identical settings on an A100-80GB GPU, our architecture reduces processing overhead at the encoder stage. As shown in Table 3, *LadderSym* achieves a significantly lower encoder latency ($\approx$97ms vs. 129ms) and improves the worst-case token latency by approximately 32ms. For a more detailed breakdown of initial processing and generation overhead, please see Table 3.

Table 3: Latency metrics for *Polytune*, *LadderSym*, and *Ladder***Bold** indicates best performance; underlined indicates second best.

| Model | Encoder latency (s) | Decoder 1st Token (s) | Worst Case Token (ms) |
|---|---|---|---|
| *Polytune* | $0.129 \pm 0.024$ | $\mathbf{0.00786 \pm 0.0356}$ | $136.86 \pm 0.0596$ |
| *LadderSym* | $\mathbf{0.0971 \pm 0.0398}$ | $\underline{0.00787 \pm 0.0201}$ | $\mathbf{104.97 \pm 0.0599}$ |
| *Ladder* | $\underline{0.0972 \pm 0.0452}$ | $0.00801 \pm 0.0364$ | $\underline{105.21 \pm 0.0816}$ |

## 4.3 ABLATIONS

In this section, we ablate two core design choices in *LadderSym*. We conduct ablations to answer the following questions: (1) How does fusion location affect performance (§4.3.1)? (2) Which input combination yields the best results in *LadderSym* (§4.3.2)?

### 4.3.1 THE EFFECT OF FUSION LOCATION ON *LadderSym*

We first examine how frequently the score and practice streams should interact inside the encoder. Starting from the *Polytune* architecture, we vary the number of joint encoder layers ($L_{\mathrm{joint}}$) while holding either the total depth or the number of modality-specific layers constant.

Table 4: *Effect of joint encoders on F1 score, measured on CocoChorales-E.* Left: Fixed total layer count to 12. Right: variant with fixed modality-specific encoders. Best results per half are in **bold**. Performance peaks at two to three joint layers, motivating our design that interleaves cross-attention while preserving modality-specific encoders.

| $L_{\text{joint}}$ | Fixed Total Layers | | | Fixed Modality Encoders | | |
|---|---|---|---|---|---|---|
| | Correct | Missed | Extra | Correct | Missed | Extra |
| 1 | 95.39% | 51.26% | 46.80% | – | – | – |
| 2 | 96.95% | **59.58%** | 57.38% | 97.00% | **59.30%** | 56.70% |
| 3 | **97.34%** | 56.81% | **59.61%** | **97.45%** | 56.14% | **57.83%** |
| 4 | 96.80% | 59.51% | 58.11% | 96.95% | 58.05% | 55.57% |
| 12 (Early Fusion) | 96.50% | 54.60% | 56.20% | – | – | – |

Table 5: *Ablations on input configurations and encoder design.* We compare prompt-only, audio-only, and combined inputs for *Polytune* and *LadderSym*, along with encoder variants that change how the two streams interact. MAESTRO-E is the more challenging dataset because of higher note concurrency, making gains there harder to realize. *LadderSym* attains the best scores on MAESTRO-E, while *Ladder* and *LadderSym* behave similarly on CocoChorales-E. Blue highlights mark *LadderSym*, pink highlights mark *Polytune*, and arrows indicate score trends; the styling is for emphasis only. These comparisons justify pairing the Ladder encoder with symbolic prompting in the final model.

| Type | Method | MAESTRO-E | | | CocoChorales-E | | |
|---|---|---|---|---|---|---|---|
| | | Missed | Extra | Correct | Missed | Extra | Correct |
| **Input Config** | Prompt Only | 24.3% | 62.5% | 90.6% | 44.6% | 45.8% | 89.4% |
| | Audio Only | 26.8% | 72.0% | 90.1% | 46.8% | 51.3% | 95.4% |
| | Prompt + Audio | 46.7% ↑ | 81.7% ↑ | 93.7% ↑ | 56.1% ↑ | 58.1% ↑ | 96.9% ↑ |
| **Encoder Design** | 3 Joint Encoders | 36.1% | 75.3% | 92.6% | 56.8% | 59.6% | 97.3% |
| | Self-Attention | 33.8% | 74.6% | 93.0% | 54.6% | 56.2% | 96.5% |
| | *Ladder* | 46.0% ↑ | 82.0% ↑ | 93.7% ↑ | 61.0% ↑ | **62.3%** ↑ | **97.8%** ↑ |
| **Final Model** | *LadderSym* | **54.7%** ↑ | **86.4%** ↑ | **94.4%** ↑ | **61.7%** ↑ | 61.4% ↓ | 97.7% ↓ |

To study fusion depth, we vary the number of joint encoders ($L_{\text{joint}}$) while keeping the total encoder layers $L_{\text{total}}$ fixed. The remaining layers are assigned to modality-specific encoders (score and practice). The number of modality-specific layers is determined by ($L_{\text{mod}} = L_{\text{total}} - L_{\text{joint}}$). As shown in Table 4, increasing $L_{\text{joint}}$ improves F1 scores. However, gains diminish beyond 2–3 joint layers (Table 4). To isolate the effect of increasing joint encoders, we also fix $L_{\text{mod}}$ and vary $L_{\text{joint}}$. This also shows peak performance at 2–3 joint encoders, followed by a decline. This confirms that too many joint encoder layers lead to diminishing returns, suggesting that there is a tradeoff between alignment capability and the ability to separately encode inputs.

### 4.3.2 ABLATION STUDY OF INPUT REPRESENTATIONS

In order to test the effectiveness of adding music scores as symbolic prompts, we evaluate three input setups: Audio Only, Prompt Only, and Audio + Prompt for both *Polytune* and *LadderSym*. Table 5 shows that Audio + Prompt outperforms both individual inputs. Symbolic prompts offer additional context for better detection. The upgraded *Ladder* encoder *improves performance on its own*, and combining it with symbolic prompting further boosts F1 scores. *LadderSym* achieves top scores overall but underperforms *Ladder* in CocoChorales-E by a small margin. We analyze this in §5.

## 5 DISCUSSION

***Analysis:*** Combining the improved encoder with the prompting strategy provides limited F1 improvements. As shown in Table 5 (*LadderSym* vs *Ladder*), this is likely because both components enhance inter-input communication in overlapping ways. The encoder improves inter-stream interaction,

while the prompt adds audio-symbolic comparison capabilities. Although the prompted version of *LadderSym* does not outperform the unprompted variant in all categories of *CocoChorales-E*, it consistently achieves better results on *MAESTRO-E*, which contains more challenging musical content (competition pieces). We therefore integrate prompts into the final version of *LadderSym*.

***Limitations:*** We identify three main limitations. First, dense concurrency present in *MAESTRO-E* acoustically masks missed notes; although *LadderSym* more than doubles missed-note F1 relative to *Polytune*, that class remains the most challenging. Second, errors accumulate near segment boundaries when sustaining notes cross the context window; those tails can be mis-labelled as extra notes, which could be mitigated with a sliding window or memory mechanism. Third, the model is designed for **fine-grained alignment** and is robust to local tempo deviations like rushed or dragged notes, but it is not intended to align performances that diverge dramatically in tempo (*e.g.*, playing at half speed). In practical tutoring settings we expect users to practice near a chosen tempo; if coarser tempo changes must be handled, a lightweight pre-alignment step can be inserted ahead of *LadderSym*.

***Future Work:*** A key direction for future work is to address the "chicken-and-egg" problem. As demonstrated in the creation of our real-world validation set (Appendix A.7), *LadderSym* can serve as a human-in-the-loop annotation tool. A larger-scale effort could use this approach to build a larger dataset of authentic performance errors, enabling the training of models that are even more robust. Beyond this, *LadderSym* introduces two key insights: cross-modal alignment should happen frequently, and asymmetric feature extraction supports better comparison. These ideas can inform reward model design or improve benchmarks for evaluating generative models. They can also guide skill assessment in other domains, such as evaluating speech or assessing physical movements.

## 6 CONCLUSION

The existing methods of music practice error detection can help more effective skill improvement, yet there remain challenges. Prior work suffers from two core drawbacks: (1) a late fusion design that restricts comparisons between practice and score streams, limiting detection F1 scores, and (2) relying on audio to represent music scores causes ambiguity. In this work, we introduced *LadderSym* to address these challenges through two key innovations: (1) a new encoder architecture featuring *alignment modules* for improved inter-stream interaction, and (2) a symbolic score prompt that reduces the ambiguity in the reference music score. This approach achieves **state-of-the-art** F1 scores on both *MAESTRO-E* and *CocoChorales-E*. On *MAESTRO-E*, it improves Missed note F1 from **26.8%** to **56.3%** and Extra note F1 from **72.0%** to **86.4%**. On *CocoChorales-E*, it improves Missed note F1 from **51.3%** to **61.7%** and Extra note F1 from **46.8%** to **61.4%**. The model's performance on our real-world beginner dataset confirms its practical utility for music learners. More broadly, this work introduces general architectural principles for sequence comparison. The insights could inform more effective models for other evaluation tasks, such as reward modeling in reinforcement learning, human skill assessment, and the benchmarking of generative models.

ACKNOWLEDGMENTS

This work was supported in part by NSF grants IIS-2326198 and OAC-2504445, the Purdue College of Engineering, and the Patti & Rusty Rueff School of Design, Art, and Performance. Any opinions, findings, and conclusions expressed in this material are those of the authors and do not necessarily reflect the views of the sponsors.

***Reproducibility Statement:*** We use the publicly available *MAESTRO-E* and *CocoChorales-E* datasets, with the creation process described in §A.6. Our newly curated real-world evaluation dataset is detailed in §A.7. Experimental Setup: Our training procedure, including all hyperparameters, is specified in §A.4 and Table 7. The evaluation metrics and statistical analysis methods are described in §4.1 and §A.11, respectively. We used a fixed seed for all experiments as documented in §A.12.

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

## A  APPENDIX

This appendix supplements the main text with additional background and experimental details. §A.1 describes our probing protocol and §A.2–A.3 detail model inputs and outputs. §A.4 outlines training settings, §A.5 defines evaluation metrics, and §A.6 covers dataset generation. §A.7 describes the real-world collection and per-piece results. §A.8 reviews the explicit-alignment baseline, §A.9 visualizes attention behaviors, §A.10 reports per-instrument results, §A.11 presents statistical tests, and §A.12 documents reproducibility practices. §A.13 lists LLM usage.

### A.1  PROBING SETUP

Probes are trained for 25 epochs on the MAESTRO-E test set. Locality is defined as predicting each token's position in a $16 \times 32$ pitch–time patch grid. Globality is measured by predicting a 12-bin energy label based on the token with the highest norm in each clip. Cross-stream correspondence is evaluated by predicting the energy bin of the opposite stream.

### A.2  MODEL INPUTS

#### A.2.1  AUDIO

Our tokenization of audio inputs follows MT3 (Gardner et al., 2022) and Polytune. We segment each audio recording into 2.145-second non-overlapping segments and compute spectrograms using the short-time Fourier transform (STFT) with a 2048-point FFT, a 128-sample hop, and 512 mel-bins. Each spectrogram frame is split into 16×16 patches using the Vision Transformer (ViT) patch embedding method (Dosovitskiy et al., 2021), yielding 512 tokens per segment for each stream.

#### A.2.2  SYMBOLIC SCORE PROMPT

As our T5 decoder is autoregressive, the symbolic score provided as an input prompt uses the same vocabulary as the model's output. The token types are described in Table 6.

Table 6: *Token types used in the symbolic score prompt and model output.*

| Token Type | Description |
|---|---|
| SOS / EOS | Start/End of Sequence. |
| Time | Specifies note timing within a segment. |
| On / Off | Indicates whether a note is played or released. |
| Label | Mistake type. In the input prompt, all notes are 'Label=Correct'. |
| Note | The MIDI pitch of the note. |

The prompt is the tokenized symbolic score. For example, a single middle C (MIDI pitch 60) played correctly at the start of a segment would be represented as follows:

```
[SOS, Time=0, Label=Correct, On, Note=60, ...]
```

### A.3  MODEL OUTPUT

Our model produces a stream of MIDI-like tokens describing each musical event's time, pitch, playback state, and error category. For example, a sequence with two errors—"extra" and "missed"—looks like:

```
[SOS, Time=0, Label=Extra, On, Note=60, Time=3, Note=60,
Off, Time=7, Label=Missed, On, Note=64, Time=9, Note=64,
Off, EOS]
```

Here, Time=0, Label=Extra starts the first erroneous note, and Time=3, Note=60, Off indicates its deactivation. Subsequent tokens (Time=7, Label=Missed, On, Note=64) mark the onset of a missed note four time steps later, followed by Off to end it. Finally, EOS

concludes the event sequence. **Repetition errors** follow the same schema. For example: if the score specifies $A \rightarrow B \rightarrow C$ but the performer plays $A \rightarrow A \rightarrow B \rightarrow C$, the second $A$ is emitted as `Label=Extra`; if they skip $B$ entirely, the timeline contains both `Label=Extra` for the duplicate $A$ and `Label=Missed` for $B$.

## A.4 TRAINING

The model is trained end-to-end using score audio, practice audio, and a symbolic score prompt. Our training recipe largely follows that of *Polytune* (Chou et al., 2025), with key adaptations to improve performance and efficiency. We apply a weighted cross-entropy loss to mitigate the class imbalance between correct and missed/extra notes. To further improve generalization, we adopt token shuffling (Tan et al., 2024), which permutes output tokens without altering underlying semantics. Learning rates are adjusted using cosine annealing (Loshchilov & Hutter, 2017), starting at $2 \times 10^{-4}$ and decaying to $1 \times 10^{-4}$. Optimization is performed with AdamW (Loshchilov & Hutter, 2017). All models are trained for 300 epochs using the largest batch size that fits on a single NVIDIA A100-80GB GPU: 48 spectrograms per batch for MAESTRO-E and 96 for CocoChorales-E. The smaller batch size for MAESTRO-E reflects its higher note density and memory footprint. Training uses mixed-precision (bf16-mixed) to balance efficiency and numerical stability. Full hyperparameters are listed in Table 7.

Table 7: *Training hyperparameters for each dataset.* Batch size refers to the number of spectrogram segments per update.

| Hyperparameter | MAESTRO-E | CocoChorales-E |
|---|---|---|
| Number of Epochs | 300 | 300 |
| Learning Rate | 2e-4 → 1e-4 (Cosine) | |
| Batch Size (spectrograms) | 48 | 96 |
| Error Loss Weight | 10 | |
| Scheduler | Cosine Annealing | |
| Optimizer | AdamW | |
| Data Augmentation | Token Shuffling | |
| Precision | bf16-mixed | |

## A.5 METRICS AND EVALUATION

Error detection metrics have varied across studies. Benetos et al. (2012); Wang et al. (2017) consider a note prediction correct if its onset falls within a specific timing tolerance relative to the ground truth. However, in this paper, we also require the pitch of the note to match, as specified by the mir_eval package (Raffel et al., 2014). Furthermore, the `mir_eval` package uses a 50 ms tolerance to calculate F1 overlap scores. In contrast, older metrics like MIREX onset accuracy employed different timing tolerances, such as 100 ms (Benetos et al., 2012) and 200 ms (Wang et al., 2017). These varying tolerances complicate direct comparisons, as higher tolerances tend to inflate accuracy scores. We use Error Detection F1 introduced by (Chou et al., 2025) because of the more stringent 50 ms tolerance from `mir_eval` and the ability to evaluate each error category separately. This provides a more precise evaluation of model performance.

## A.6 TRAINING DATASETS

Training an end-to-end model for music error detection requires a large volume of labeled performance mistakes. Yet, no large-scale datasets are available for this task. The only prior dataset, introduced by Benetos *et al.* (Benetos et al., 2012), contains just 7 tracks. To address this limitation, Chou et al. (2025) developed two new datasets: *MAESTRO-E* and *CocoChorales-E*, each containing over 1,000 samples per instrument. *MAESTRO-E* provides more than 200 hours of piano audio across 1,000+ tracks, annotated with over 200k pitch and timing errors. *CocoChorales-E* spans 300+ hours of audio with over 40,000 tracks and 13 instruments, capturing more than 25,000 annotated errors. In contrast, the dataset from Benetos *et al.* includes only 15 minutes of audio, 7 tracks, and 40 labeled

errors. To generate these datasets, MIDI samples from the MAESTRO and CocoChorales corpora were augmented with typical practice mistakes such as missed, incorrect, and additional notes. The corresponding audio was synthesized using MIDI-DDSP (Wu et al., 2022).

Training labels were defined by segmenting each augmented MIDI file into three separate MIDI tracks labeled as *Correct*, *Missed*, and *Extra*, following the definitions introduced in §1.

---

**Algorithm 1** *MIDI error generation algorithm.* This procedure introduces missed notes, pitch changes, timing shifts, and extra notes into a clean MIDI file. Reproduced from Chou et al. (2025).

---

**Require:** All notes in MIDI track $A$, error rate $\lambda$, offset distributions $P, Q$.
 1: Select notes from $A$ to augment with probability $\lambda$
 2: **for** each note selected **do**
 3:     err_type $\leftarrow$ rand( {missed note, pitch change, timing shift, extra note} )
 4:     **if** err_type = missed note **then**
 5:         Remove note;
 6:     **else if** err_type = pitch change **then**
 7:         $\epsilon_p \leftarrow$ sample($P$)
 8:         Offset pitch by $\epsilon_p$;
 9:     **else if** err_type = timing shift **then**
10:         $\epsilon_t \leftarrow$ sample($Q$)
11:         Offset time by $\epsilon_t$;
12:     **else if** err_type = extra note **then**
13:         $\epsilon_p \leftarrow$ sample($P$)
14:         $\epsilon_t \leftarrow$ sample($Q$)
15:         Insert note with time offset $\epsilon_t$ and pitch offset $\epsilon_p$;
16:     **end if**
17: **end for**

---

The error injection process is outlined in Algorithm 1. Notes from a MIDI track are randomly selected with a probability determined by $\lambda$, which is sampled from a uniform distribution $U(0.1, 0.4)$. Each selected note is then assigned an error type. Depending on the error, the note is removed, modified in pitch or timing, or a new note is inserted with sampled pitch and timing offsets. The offsets are drawn from two truncated normal distributions $P$ and $Q$, centered at zero with standard deviations of 1 and 0.02, respectively. These distributions are chosen to reflect realistic variations observed in human performance(Trommershäuser et al., 2005; Tibshirani et al., 2011). An overview of the resulting datasets is presented in Table 8.

Table 8: *Key properties of music practice error-detection datasets. Each track contains multiple missed or extra note errors and randomly timed timing perturbations.*

| Dataset | Tracks | Instruments | Errors | Source |
|---|---|---|---|---|
| MAESTRO-E | 1k+ | Piano | 200k+ | Partially-Synthetic |
| CocoChorales-E | 40k+ | 13 | 25k+ | Partially-Synthetic |
| Benetos et al. | 7 | Piano | 40 | Professional |
| Our real-world dataset | 10 | Piano | 161 | Beginners |

## A.7 REAL-WORLD EVALUATION DATASET

This section documents the out-of-distribution benchmark we curated to complement the partially synthetic corpora. Table 9 summarizes per-piece F1 scores, and Appendices A.7.1 to A.7.3 describe how we recorded the performances, annotated them, and assessed model behavior on this set.

### A.7.1 DESCRIPTION OF REAL-WORLD DATASET

While large-scale synthetic data is crucial for training, a key limitation of prior work is the reliance on synthetic or scripted mistakes for evaluation, which may not capture the full complexity and

nuance of real human performance. Collecting a dataset of authentic errors is a time-consuming and challenging task. Unlike generating synthetic data or pretending to make a mistake, it requires capturing genuine mistakes made by musicians during the learning process. This involves finding beginner musicians and allowing them to practice on the spot, a process necessary to record the natural, unscripted errors. Furthermore, annotating these errors is even more difficult, as it requires a fairly well-trained ear to discern subtle inaccuracies. This need for domain expertise makes the task ill-suited for general crowdsourcing platforms.

To complement the partially-synthetic benchmarks, we captured a held-out corpus of beginner performances recorded on the piano by beginners (statistics shown in Table 8). The dataset was acquired by capturing the direct audio output from a digital piano setup, which utilized a VST (sampled from a Yamaha Disklavier). This method ensures a clean signal and consistent recording conditions. Each take includes the reference MIDI score, practice audio, and manual note-level labels. Because some pianos offers symbolic MIDI ground truth with well-defined onsets, annotators could compare MIDI labels rather than transcribe every event purely by ear before comparing, keeping dataset labeling efforts feasible at a small scale. The ground truth annotations contain *75 wrong-note pairs* (a substitution of one note for another), *51 extra notes*, and *35 missed notes*, making isolated missed notes the rarest error category. To our knowledge, this is the largest publicly available dataset of real-world, annotated beginner performances curated specifically for music error detection, representing an almost threefold increase over previously existing real-data benchmarks. This expansion provides the community with a much-needed resource to validate models against real-world data.

Recording sessions featured three beginner pianists: one outside the author list who studied piano for only a year in childhood, one coauthor with no formal training, and one who plays guitar but has minimal keyboard experience. We curated twenty popular beginner–easy pieces and asked performers to play on the spot with minimal rehearsal so that mistakes arose naturally. Labels were produced by a author together with another student classically trained in sight-reading.

Assembling this dataset was far from trivial. Each participant recorded six to seven pieces under supervision, yielding a total of 12 hours of recording and practicing time, and annotation required multiple passes by musically trained reviewers to verify every note-level label, taking up to approximately 52 person hours to annotate. Scaling beyond 20 pieces would require hundreds of hours of additional expert time, so the dataset remains modest in absolute size. Crowdsourcing a comparable resource remains an open problem: most potential contributors lack access to recording setups and sufficient musicianship to label fine-grained mistakes. Yet, to our knowledge, this corpus is the largest publicly available set of authentic beginner errors, and it provides a valuable resource to assessing whether models trained on synthetic data generalize to real practice.

### A.7.2 LADDERSYM AS AN ANNOTATOR

To accelerate the laborious process of manual annotation, we use *LadderSym* as an assistive labeling tool. The model generates a first pass of annotations which, while not perfect, provides a strong starting point. These preliminary labels, including any errors made by the model, are then meticulously reviewed, corrected, and verified by two human annotators to establish the final ground truth. This "human-in-the-loop" approach is significantly faster than labeling from scratch. By shifting the task from pure annotation to verification and correction, we estimate that this process cut the required labeling time in half, speeding up throughput by $3\times$.

### A.7.3 EVALUATION RESULTS

We evaluate all models on this dataset without additional finetuning. On this real-data set, extra notes are generally easier to detect as they stand out more in these simpler beginner pieces compared to the dense, complex arrangements in a competition dataset like MAESTRO-E. Despite the challenges of real-world data, the relative performance ordering observed on synthetic data persists. *LadderSym* shows a significant improvement in detecting the more challenging missed notes and also demonstrates a modest improvement in extra note detection over *Polytune*, highlighting its robustness. Per-piece metrics are shown in Table 9.

Table 9: *Per-piece F1 comparison on the real-data evaluation set.* All models are trained solely on the synthetic datasets. Values are percentages; "–" indicates the metric is undefined because the practice recording contains no events of that type.

| Piece | LadderSym | | Polytune | |
|---|---|---|---|---|
| | Extra | Missed | Extra | Missed |
| Amazing Grace | 92.3 | 76.9 | 100.0 | 76.9 |
| Für Elise | 80.0 | 88.9 | 62.5 | 53.3 |
| Greensleeves | 83.3 | 72.7 | 90.9 | 60.0 |
| Morning (Grieg) | 100.0 | 75.0 | 100.0 | 44.4 |
| Happy Birthday | 100.0 | 66.7 | 100.0 | 66.7 |
| Jupiter (Holst) | 80.0 | 66.7 | 80.0 | 0.0 |
| Jingle Bells | 71.4 | 60.9 | 76.9 | 47.6 |
| London Bridge | 66.7 | – | 10.5 | – |
| Mary Had a Little Lamb | 100.0 | 66.7 | 100.0 | 53.3 |
| Merry Christmas | 100.0 | 100.0 | 88.9 | 88.9 |
| Can-Can (Offenbach) | 80.0 | 100.0 | 100.0 | 66.7 |
| Row, Row, Row Your Boat | 92.3 | 76.9 | 83.3 | 66.7 |
| Scale Exercise | 100.0 | 100.0 | 100.0 | 100.0 |
| Silent Night | 45.5 | 30.0 | 66.7 | 8.0 |
| Surprise (Haydn) | 83.7 | 88.2 | 77.3 | 85.7 |
| Twinkle Twinkle Little Star | 85.7 | 100.0 | 100.0 | 100.0 |
| When the Saints Go Marching In | 80.0 | 76.2 | 88.9 | 82.4 |
| Hot Cross Buns | 66.7 | – | 40.0 | – |
| Itsy Bitsy Spider | 66.7 | 100.0 | 66.7 | 50.0 |
| Old McDonald Had a Farm | 57.1 | 66.7 | 80.0 | 100.0 |
| **Average** | **81.6** | **78.5** | **80.6** | **63.9** |

## A.8 EXPLICIT-ALIGNMENT BASELINE

We adopt the same baseline introduced by Chou et al. (2025). Their work provides an updated, open-source implementation of the methods by Benetos *et al.* and Wang *et al.*, which remain the most directly relevant to score-informed error detection (Benetos et al., 2012; Wang et al., 2017). While preserving the core principles of the original approaches, the re-implementation modify each stage of the transcription pipeline to align with recent progress in automatic music transcription (AMT). In particular, they replace the non-negative matrix factorization (NMF)-based transcription with the MT3 (Gardner et al., 2022) model, a state-of-the-art system. They also substitute Windowed Time Warping (WTW) with the more accurate Dynamic Time Warping (DTW). These updates yield comparable performance while extending support to multi-instrument settings.

## A.9 ATTENTION PATTERN VISUALIZATION

We compare attention behaviors of three encoder designs: early fusion, late fusion (*Polytune*), and our proposed cross-attention alignment module (*LadderSym*).

### A.9.1 POLYTUNE SELF-ATTENTION

Figure 6 visualizes per-layer attention maps from POLYTUNE. Diagonal patterns dominate the practice stream, while the practice stream exhibits vertical banding, indicating reduced temporal specificity. This asymmetry reflects that the practice stream encodes more global structure, while the practice stream retains local detail. The final layer also exhibits vertical banding, showing lack of locality in one of the streams.

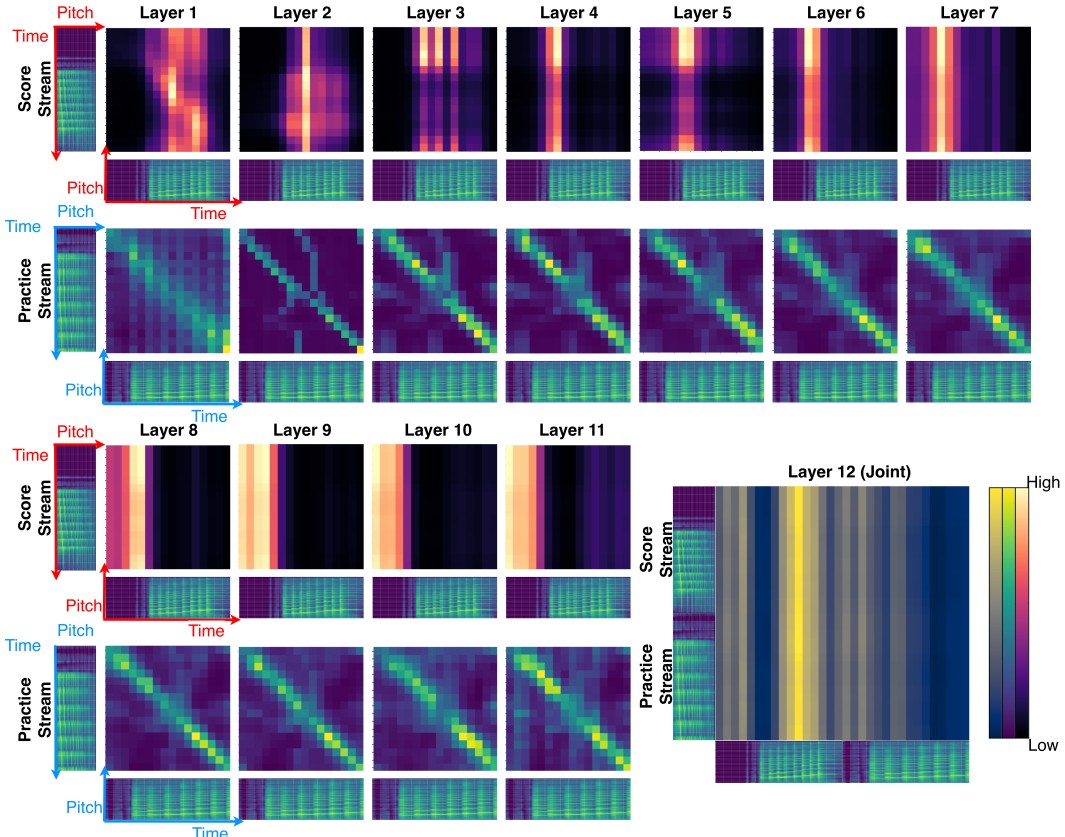

Figure 6: *Per-layer attention maps in* POLYTUNE. Maps are averaged over pitch. Layers 1–11 use independent encoders; layer 12 uses a joint encoder. The score stream (top row) loses its local, diagonal attention structure in deeper layers. This demonstrates that this stream is sacrificing local information to build a compressed, global representation of the entire clip. As tokens pass through deeper layers, the model can progressively 'mix' with information from other tokens, thus causing the tokens to become less tied to their initial spatial location.

### A.9.2 EARLY FUSION SELF-ATTENTION (FULLY JOINT)

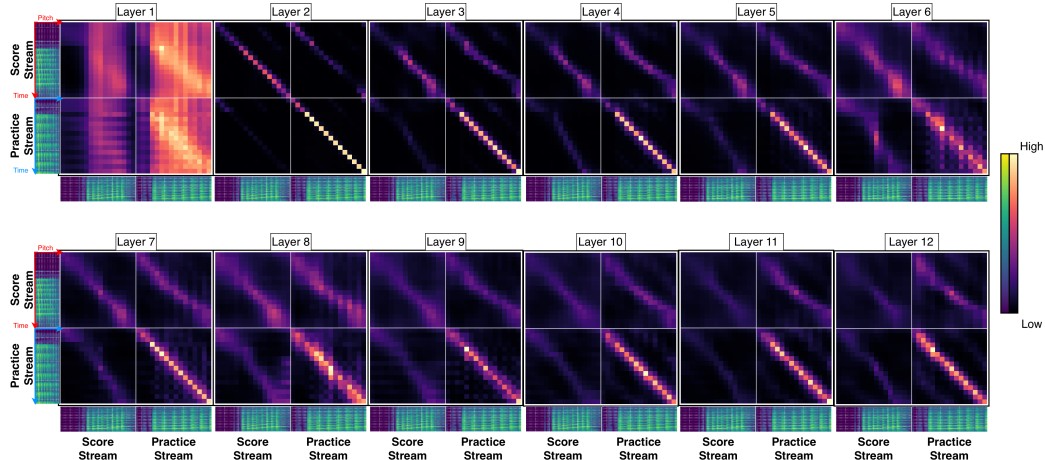

Figure 7: *Self-attention maps for the early fusion model.* Each quadrant shows intra- or inter-stream attention, averaged over the pitch axis. We observe strong alignment, but also strong locality in the intra-stream attention.

Figure 7 shows quadrant attention maps from the FULLY JOINT early fusion encoder. Each quadrant represents one attention pattern: top-left is practice-to-practice, bottom-right is practice-to-practice, and the off-diagonal quadrants capture practice-to-practice and practice-to-practice attention. All maps are averaged over the pitch axis to emphasize temporal alignment. This encoder exhibits strong diagonal structures, indicating that tokens attend mostly to themselves or nearby frames, preserving strong local correspondence in both streams.

### A.9.3 LADDERSYM CROSS-ATTENTION

Figure 8 illustrates the cross-attention maps in *LadderSym*. These maps are averaged over pitch to highlight temporal alignment. Unlike the previous models, *LadderSym* inserts cross-attention modules at each layer, enabling continuous alignment between the practice and practice streams. Earlier layers show more distinct diagonals, while later layers shift toward abstract correspondence. Moreover, we show that asymmetry is preserved via probing in Table 1: one stream remains locally detailed while the other emphasizes cross-stream integration. We further investigated the first-layer

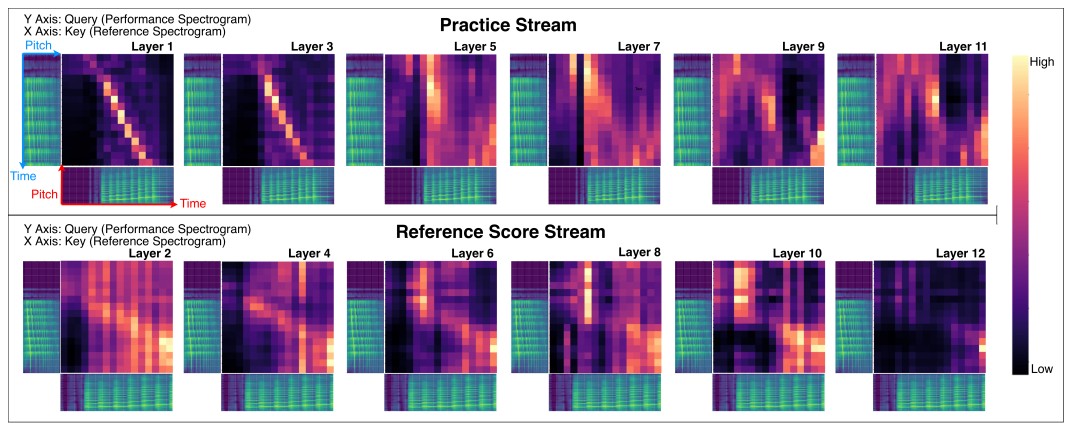

Figure 8: *Cross-attention maps in LadderSym, averaged over pitch.* Axes represent token positions in practice and practice streams.

attention maps for specific error topologies to determine if interpretable geometric patterns, such as gaps for omissions or cross patterns for reversals, emerge. As shown in Figure 9, distinct these geometric variations are not observable in the first layer.

We hypothesize that more abstract error-type reasoning occurs in deeper layers. In the case of skipped notes, the absence of a "gap" is likely because the model must actively attend to the corresponding silence in the audio stream to verify the absence of the expected pitch.

Similarly, the absence of cross-patterns in "flipped" sequences is a result of our error taxonomy. We treat a permuted sequence not as a distinct structural class, but as a combination of insertion and deletion errors. For instance, if a score sequence $A \to B \to C$ is performed as $C \to B \to A$, the model resolves this locally as a "Missing $A$ / Extra $C$" event at the first position and a "Missing $C$ / Extra $A$" event at the last, rather than actually detecting a reversal. Consequently, there is no incentive to learn a cross pattern.

### A.10 INSTRUMENT-LEVEL RESULTS

To expose the variation across instruments, we report per-instrument Error Detection F1 scores for *LadderSym*, *Polytune*, and the explicit-alignment baseline.

We present instrument-level results of *LadderSym* versus prior work for every instrument in Table 10. We also provide qualitative examples in our demo of the violin, piano, flute, and tenor sax.

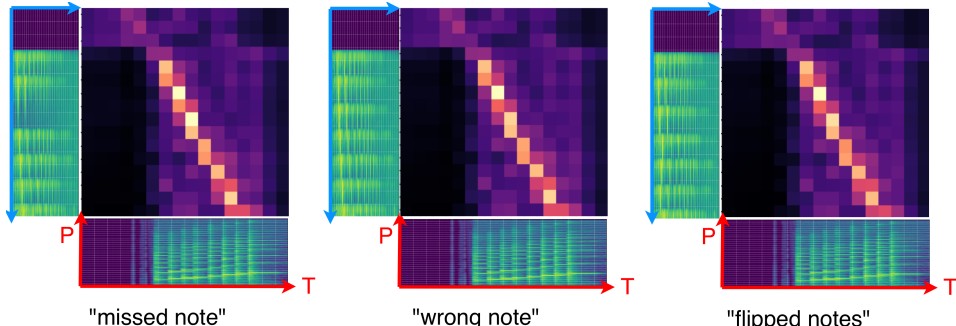

"missed note"        "wrong note"        "flipped notes"

Figure 9: *First layer cross-attention maps in LadderSym, for various error scenarios.* Axes represent token positions in practice and practice streams. Despite the differing error types, the attention patterns at this initial layer exhibit highly similar, strong diagonal alignments, indicating that distinct signs of error detection are not yet observable at this stage and likely happen in later layers.

Table 10: Full results of error detection F1 scores for 14 instruments, split into Correct, Missed, and Extra note categories. We compare three models (*LadderSym*, Polytune, and a baseline). The row labeled "Average" summarizes all 14 instruments: piano from *MAESTRO-E* plus 13 additional instruments from *CocoChorales-E*. *LadderSym* has better F1 scores across the board compared to other methods.

| Instrument | Correct (F1) | | | Missed (F1) | | | Extra (F1) | | |
|---|---|---|---|---|---|---|---|---|---|
| | **Ours** | **Polytune** | **Baseline** | **Ours** | **Polytune** | **Baseline** | **Ours** | **Polytune** | **Baseline** |
| **Average** | **97.4%** | 95.0% | 37.0% | **61.2%** | 49.2% | 7.6% | **63.2%** | 48.0% | 25.9% |
| **Piano** | **94.4%** | 90.1% | 43.5% | **54.7%** | 26.8% | 6.6% | **86.4%** | 72.0% | 39.9% |
| **Flute** | **97.9%** | 96.0% | 38.9% | **68.7%** | 56.0% | 7.2% | **65.0%** | 52.0% | 26.6% |
| **Clarinet** | **97.8%** | 95.6% | 38.3% | **59.0%** | 49.7% | 6.7% | **61.0%** | 46.6% | 24.1% |
| **Oboe** | **98.0%** | 96.3% | 33.4% | **69.9%** | 58.4% | 6.7% | **62.6%** | 48.1% | 25.9% |
| **Bassoon** | **97.6%** | 94.4% | 34.7% | **62.2%** | 48.9% | 6.4% | **62.7%** | 41.7% | 17.1% |
| **Violin** | **97.6%** | 95.5% | 36.1% | **68.2%** | 57.1% | 7.5% | **62.9%** | 48.8% | 27.3% |
| **Viola** | **97.6%** | 95.1% | 36.1% | **57.2%** | 46.9% | 5.9% | **59.9%** | 47.7% | 26.1% |
| **Cello** | **97.7%** | 94.9% | 37.5% | **52.6%** | 42.7% | 6.9% | **61.4%** | 46.8% | 21.7% |
| **Trumpet** | **98.1%** | 96.3% | 37.8% | **65.6%** | 58.7% | 8.8% | **65.3%** | 53.6% | 26.6% |
| **French Horn** | **97.8%** | 96.1% | 38.4% | **61.8%** | 53.9% | 5.9% | **57.1%** | 43.2% | 23.7% |
| **Tuba** | **97.7%** | 95.2% | 37.3% | **55.9%** | 45.4% | 8.1% | **64.8%** | 45.6% | 17.8% |
| **Trombone** | **96.8%** | 94.8% | 35.0% | **59.8%** | 50.4% | 7.1% | **58.7%** | 44.8% | 21.7% |
| **Contrabass** | **97.5%** | 94.2% | 35.7% | **54.9%** | 42.0% | 8.9% | **56.6%** | 38.6% | 19.9% |
| **Tenor Sax** | **98.6%** | 95.7% | 39.7% | **66.9%** | 56.2% | 14.2% | **60.4%** | 45.7% | 25.1% |

## A.11 STATISTICAL ANALYSIS

To assess significance across the explicit-alignment baseline, *Polytune*, and *LadderSym*, we ran paired $t$–tests and Wilcoxon signed-rank tests on CocoChorales-E and MAESTRO-E (Bonferroni-corrected $\alpha = 0.017$), and show the results in Table 11. Some of the computed $p$-values were smaller than the smallest magnitude reliably distinguishable from zero in standard double precision ($\approx 10^{-308}$), so we report them as $< 1 \times 10^{-300}$. Even at this threshold, all $p$-values remain far below our significance level.

## A.12 SEED MANAGEMENT FOR REPRODUCIBILITY

To ensure reproducibility, we implemented a consistent seed management strategy for model training. We utilized specific seeds for each stage to ensure that results could be replicated exactly. **Model Training:** For model training, we used PyTorch Lightning's `seed_everything` function with a seed value of 365. This seed was applied across all relevant components of the training process,

Table 11: *Paired t–test and Wilcoxon signed-rank results.*

| Dataset | Comparison | $t$ | $p_t$ | $W$ | $p_w$ |
|---|---|---|---|---|---|
| **CocoChorales-E** | *Polytune* vs. Explicit Align. | 106.98 | $< 1 \times 10^{-300}$ | $1.38 \times 10^6$ | $< 1 \times 10^{-300}$ |
| | Explicit Align. vs. *LadderSym* | -127.10 | $< 1 \times 10^{-300}$ | $8.75 \times 10^5$ | $< 1 \times 10^{-300}$ |
| | *Polytune* vs. *LadderSym* | -21.85 | $4.98 \times 10^{-103}$ | $1.79 \times 10^6$ | $1.47 \times 10^{-170}$ |
| **MAESTRO-E** | *Polytune* vs. Explicit Align. | 86.18 | $< 1 \times 10^{-300}$ | $2.28 \times 10^4$ | $< 1 \times 10^{-300}$ |
| | Explicit Align. vs. *LadderSym* | -110.31 | $< 1 \times 10^{-300}$ | $6.87 \times 10^3$ | $< 1 \times 10^{-300}$ |
| | *Polytune* vs. *LadderSym* | -20.53 | $1.43 \times 10^{-85}$ | $6.25 \times 10^5$ | $1.03 \times 10^{-67}$ |

including data loading, model initialization, and training loops, to ensure that training is consistent and reproducible across different runs. The following code snippet (Listing 1) demonstrates how the seed was set for model training:

Listing 1: Setting a seed with PyTorch Lightning's seed_everything

```python
from pytorch_lightning import seed_everything

# Set seed for model training
seed = 365
seed_everything(seed)
```

### A.13    LLM USE IN MANUSCRIPT PREPARATION

We used OpenAI's GPT-5 (via ChatGPT) to help with wording clarity and grammar-only edits. All scientific claims, experimental design, data analysis, and conclusions remain the responsibility of the authors.

