# OpenReview forum: "LadderSym: A Multimodal Interleaved Transformer for Music Practice Error Detection"
_ICLR.cc/2026/Conference — ICLR 2026 Poster_

### Official Review · Reviewer_3opm · 2025-10-29

**Soundness:** 3
**Presentation:** 3
**Contribution:** 3
**Rating:** 8
**Confidence:** 3

**Summary:**

LadderSym introduces an interleaved two-stream transformer for music practice error detection.
By inserting cross-attention alignment modules and symbolic score prompts, it achieves substantial improvements over Polytune and classic DTW-based baselines. The model detects missed, extra, and wrong notes with high precision.

**Strengths:**

1. Clear empirical improvement with solid ablation and baseline comparison. Music error detection is, in general, a challenging problem without elegant solutions so far. Very glad to see such a contribution.

2. A fusion design that enhances alignment while preserving modality specialization.

3. A connection with a larger picture: the model presented is, in spirit, well resonant with "function alignment"(https://arxiv.org/abs/2503.21106), a new theory of mind that is also closely related to music perception. I recommend citing and relating to it.

**Weaknesses:**

Possible improvements:

The paper could emphasize what is newly learned: attention visualization would clarify whether the model develops interpretable alignment behaviors.

The method could potentially apply to other music performance problems  (say, expressive performance).

**Questions:**

Is the demo in real time? It looks like so from the demo page, but the method in the paper does not seem to support a real-time function yet.

---

> ### Author Response · Authors · 2025-11-20
>
> We thank the reviewer for their review and appreciate their insightful feedback. Below, we address overall concerns and questions:
>
> ### Weakness/Possible improvements
>
> > The paper could emphasize what is newly learned: attention visualization would clarify whether the model develops interpretable alignment behaviors.
>
> Perhaps the reviewer missed our treatment of this topic. Attention visualizations can be found in Figure 3, which visualizes the learned alignment in our first-layer module, showing how it learns a meaningful temporal correspondence and recovers the same off-diagonal structure as DTW alignments. Additionally, Appendix A.10.3 (Figure 8) provides the full cross-attention maps for all layers of LadderSym, showing how the alignment shifts from distinct diagonals in earlier layers to more abstract correspondence in later ones. We will add a sentence to the main text to more explicitly connect these figures and probing results to the concept of interpretable learned alignment.
>
> > The method could potentially apply to other music performance problems (say, expressive performance).
>
> This is an exciting direction for future work. We believe the reviewer is referring to applying our comparison architecture to analyze expressive performance (e.g., evaluating micro-timing, dynamics, or articulation) against a reference score, which is an extension of our work we are researching. We agree that our LadderSym architecture, as a general-purpose model for comparing two related sequences, could be effective for this. Training data incorporating timing and dynamics could be obtained by extending the synthetic data generation from Polytune.
>
> Other aspects of expressive performance, like articulation, is up to human interpretation and does not have strict rubric to define a clear ground truth. One way to evaluate articulation is to compare to a reference ground truth and have a model label the discrepancy. However, since labels on existing performances are difficult to obtain, a probable solution that we plan to look into in the future is editing existing performances to modify the articulations. Since we can define the magnitude of the edits, the model will have labels for the discrepancy between the reference and performance. This approach could apply to micro-timing and dynamics as well. However, we believe better controllability in editing models is needed to obtain the training data for this goal. Current editing models still struggle with consistency and are mostly still focused on audio level edits rather than note-level edits. Thus, we leave this for future work.
>
>
> ### Questions
> > A connection with a larger picture: the model presented is, in spirit, well resonant with "function alignment"(https://arxiv.org/abs/2503.21106), a new theory of mind that is also closely related to music perception. I recommend citing and relating to it.
>
> Thank you for this recommendation. We were not aware of this recent work, but we agree that it provides an interesting theoretical lens through which to interpret our findings. Our own work was inspired by how a human could process error detection, by comparing a low-level representation (the heard audio) with a high-level one (the remembered score or audio). This intuitive model of comparing a low-level stream to a high-level stream maps well to the x (low-level) and z (high-level) streams in the "Function Alignment" paper.
>
> The "Function Alignment" paper offers a compelling theoretical interpretation of our results. For example, our probing analysis in Section 3.1 (Probing Ladder) and Table 1, which found that our Ladder architecture leads to an "asymmetric division of labor," could be seen as an example of the "mutual adaptation" described in that paper. We will cite and discuss this paper in our final version as a way to interpret our architectural findings.
>
>
> > Is the demo in real time? It looks like so from the demo page, but the method in the paper does not seem to support a real-time function yet.
>
> The demo video shows a pre-recorded performance being processed. The model's measured latency is sufficient for real-time applications. Please see our response to reviewer gi3B on system inference latency for more details.

---

> > ### Comment · Reviewer_3opm · 2025-11-27
> >
> > The attention visualizations indeed clarify that LadderSym learns DTW-like temporal correspondence, which is helpful for alignment interpretability.
> >
> > It would be interesting, however, to further explore error-type interpretability: whether distinct attention patterns correspond to specific performance errors such as skips, reversals, or pitch mismatches.
> >
> > For example, can a skip manifest as a diagonal gap, or a reversal as a cross-pattern in the attention map?
> > Demonstrating such diagnostic interpretability would make the model more insightful for music performance analysis beyond alignment accuracy.
> >
> > The score remains -- good paper.

---

> > > ### Author Response · Authors · 2025-12-03
> > >
> > > We further investigated the first-layer attention maps for specific error topologies to determine if interpretable geometric patterns, such as gaps for omissions or cross patterns for reversals, emerge.  As
> > > now shown in Figure 9 of the updated manuscript, distinct these geometric variations are not observable in the first layer.
> > > We hypothesize that for skipped notes, the model must actively attend to the audio gap to verify absence; if it ignored that region, it would lack the context to predict a "missing" label. Similarly, because our objective treats "flipped" sequences as combinations of missing and extra notes rather than a distinct "reversal" class, the model has no incentive to learn a "cross-pattern" mechanism at the alignment level. For instance, if a score sequence A→B→C is played as C→B→A, the first position is labeled as "Missing A / Extra C" and the last position as "Missing C / Extra A". We suspect that specific error-type reasoning occurs in later layers, but this is harder to demonstrate conclusively. Unlike single-stream models, where approximations like attention rollout can easily trace input contributions, our architecture continuously mixes representations between streams. Consequently, the relation to the initial positions is lost in deeper layers, as the continuous mixing means there is no longer a single, traceable path from the output back to the input.

---

### Official Review · Reviewer_BK2b · 2025-10-29

**Soundness:** 2
**Presentation:** 2
**Contribution:** 2
**Rating:** 4
**Confidence:** 3

**Summary:**

This paper introduces *LadderSym* for music error detection, featuring (1) a two-stream encoder with inter-stream alignment modules, and (2) the integration of both audio and symbolic scores as reference inputs. The model achieves state-of-the-art performance on both the *MAESTRO-E* and *CocoChorales-E* datasets.

**Strengths:**

1. The improvement over the previous model (*Polytune*) is substantial.
2. The proposed *Ladder* structure could potentially influence other tasks that require comparison between two sequences.
3. The analysis of fusion stages provides useful insights for designing other multi-input models.
4. Experiments conducted on a curated real-world dataset add value; if released publicly, this dataset could have a meaningful impact on piano error detection research.

**Weaknesses:**

1. The effect of incorporating symbolic scores is not sufficiently strong, as the relatively small gap shown in Table 4 weakens one of the paper’s two main contributions.
2. The contribution of the *Ladder* design itself is questionable. Based on Tables 3 and 4, the position of the fusion stage appears to have a greater influence than the Ladder architecture, which fuses at every layer. On the *CocoChorales-E* dataset, simply moving from late fusion (*Polytune*) to the last three fusion layers improves the Extra-note F1 score from 46.8% to 59.6%. Introducing Ladder further raises it only slightly, to 62.3%. This suggests that the fusion location, rather than the Ladder mechanism, drives the improvement, making the claim that “cross-modal alignment should happen frequently” less convincing.
3. The writing could be improved. Figures and tables are scattered throughout the text, and Section 3 (*Method*) is intertwined with experimental results. It is recommended to separate methodological and experimental sections more clearly for better readability.

**Questions:**

Could the authors please address the following:

1. What does *globality* refer to, and why is coarse clip-level energy information used to measure it?
2. Why is asymmetric feature extraction important for error detection? Intuitively, both practice and score streams should capture local information to localize and classify erroneous notes.
3. Given the additional fusion steps, why does *LadderSym* have fewer parameters and faster inference speed compared to *Polytune*?
4. In Figure 6, for the score stream, why are the attention maps concentrated at the beginning, especially in the last few layers (8–11), when the corresponding horizontal frames appear almost silent?

---

> ### Author Response · Authors · 2025-11-20
>
> We thank the reviewer for the detailed review and are grateful for your critical feedback. Below, we address overall concerns and questions:
>
> ### Weaknesses
>
> > The effect of incorporating symbolic scores is not sufficiently strong...
>
> We respectfully disagree that this contribution is weak. The reviewer notes the "relatively small gap," but we believe the data in Table 4 (Encoder Design) shows a meaningful improvement on both our contributions.
>
> To isolate the effect of prompting (our second contribution), we show in Table 4 that prompting improves missed note detection by **19.9 F1 points** for MAESTRO-E and **9.3 F1 points** for CocoChorales-E over prior SOTA (Prompt+audio vs Audio only).
>
> The ladder encoder further improves performance on the already complex, polyphonic MAESTRO-E dataset. Missed-note F1 improves from 46.0% to 54.7% (+8.7 points), and Extra-note F1 improves from 82.0% to 86.4% (+4.4 points). Given that missed notes were the most challenging error type for prior work, we believe that this gain in isolation (an 8.7-point absolute improvement, or 19% relative gain) is a significant contribution, even in other fields of deep learning work.
>
> > the position of the fusion stage appears to have a greater influence...
>
> To clarify, the improvement from fusion location (e.g., "3 Joint Encoders") is a finding we present in Table 3 to demonstrate the limitations of simple fusion. We also report in Table 3 that performance gains diminish beyond 2-3 joint layers, demonstrating that the number of joint encoders requires tuning. Our analysis in Section 3.1 suggests this performance drop occurs because fusing too early sacrifices cross-stream feature extraction due to parameter sharing, which limits the streams' ability to specialize.
>
> The Ladder architecture is our solution to this trade-off, enabling alignment (or fusion) at every layer without forcing parameter sharing. As Table 4 (Encoder Design) clearly shows, our "Ladder" model outperforms the best-tuned baseline ("3 Joint Encoders") on the challenging MAESTRO-E dataset (e.g., 46.0% vs. 36.1% on Missed-note F1).
>
> > Writing improvements
>
> Thank you for the suggestion. We will make sure to update the table locations to improve flow.
>
> ### Questions
>
> > What does globality refer to, and why is coarse clip-level energy information used to measure it?
>
> The term “globality” is defined in our probing analysis in Section 3.1 and Appendix A.2. In this context, "globality" refers to clip-level features (information that could only be inferred from seeing the whole clip). We used "coarse clip-level energy" as a simple, easy-to-compute proxy to test our hypothesis about whether the streams were specializing. Other proxies may work, but we chose this because it is application-independent (compared to something like the number of notes in a frame).
>
> > Why is asymmetric feature extraction important for error detection?
>
> Our results in Table 1 show that both streams still capture local information, but capturing global information, especially across streams, is more important. Asymmetric feature extraction is analogous on a high level to classical methods like DTW, which also compare one local point in a sequence to the entirety of the other sequence to find an optimal path. We found the model learns to use one stream as a more "global" reference or context (e.g., the score), while the other stream (e.g., the practice) performs a more "local" search to find alignments and deviations against that context. Figure 3 explicitly shows how our alignment module's learned attention maps recover the same off-diagonal structure as DTW alignments.
>
> > on LadderSym parameters and inference speed compared to Polytune
>
> This is an advantage of our design. Polytune uses two separate 11-layer encoders plus a 1-layer joint encoder, for a total of 23 transformer blocks. In contrast, LadderSym uses 6 layers per stream, and these are interleaved to create a depth of 12 layers. Although our LadderSym adds cross-attention alignment modules, these have less parameters than Polytune’s full transformer block, making the overall design more lightweight (172M vs. 192M) and faster (by ~20%).
>
> > In Figure 6, for the score stream, why are the attention maps concentrated at the beginning, especially in the last few layers (8–11), when the corresponding horizontal frames appear almost silent?
>
> This is an excellent observation of Figure 6! The model is not focusing on silence. Based on our probing results, these tokens are sacrificing local information to store global information. We have clarified this in the caption of the figure. Essentially, the score stream is sacrificing local information to build a compressed, global representation of the entire clip. As tokens pass through deeper layers, the model can progressively 'mix' with information from other tokens,  causing the tokens to become less tied to their initial spatial location.

---

### Official Review · Reviewer_bM19 · 2025-11-01

**Soundness:** 2
**Presentation:** 3
**Contribution:** 2
**Rating:** 4
**Confidence:** 3

**Summary:**

The paper introduces LadderSym, a transformer-based model that improves error detection in music practice through two innovations:
1)	The paper proposes a two-stream “Ladder” encoder that processes the practice audio and reference audio separately but aligns them at each layer using cross-attention (“inter-stream alignment modules”), which improves synchronization and comparison without forcing both streams to share parameters.
2)	Symbolic prompting, a prompting method that adds a symbolic (MIDI) score as a decoder prompt alongside the audio input, which reduces ambiguity caused by overlapping frequencies in the score’s audio.
These components together allow more accurate note-level classification of mistakes.

**Strengths:**

The system utilizes inter-stream alignment in every encoder layer, enabling the model to learn temporal dependencies similar to classical DTW but more robustly. The symbolic prompt adds contextual knowledge that helps with subtle or overlapping notes. Probing experiments show that each stream learns to specialize.

**Weaknesses:**

The model itself still struggles towards dense musical passages which might mask missed notes. Some errors near sequence boundaries may be mislabeled. The model is proposed under the assumption of roughly stable tempo, and extreme tempo deviations still require pre-alignment.

**Questions:**

1.	The authors mention “20 beginner performances” but do not specify: a) The number of unique players (only “three graduate students”) vs. performances (20 total — overlap unclear). b) The recording conditions (microphone type, room noise, instrument consistency).
2.	How long are the symbolic prompts relative to the output sequence (do they truncate or pad)? Does the model ever ignore the prompt when symbolic and audio information disagree?
3.	Was any ablation done on partial or noisy symbolic prompts (realistic for imperfect MIDI scores)?

**Details Of Ethics Concerns:**

No.

---

> ### Author Response · Authors · 2025-11-20
>
> We thank the reviewer for the detailed review and are grateful for your critical feedback. Below, we address overall concerns and questions:
>
> ### Weaknesses
>
> > The model itself still struggles towards dense musical passages, which might mask missed notes. Some errors near sequence boundaries may be mislabeled. The model is proposed under the assumption of a roughly stable tempo, and extreme tempo deviations still require pre-alignment.
>
> These are indeed challenges that our model struggles with. We elaborate on each point below.
>
> **Dense concurrency:**
> Although dense concurrency causes missed notes to remain the most challenging error class, our work describes an over two-fold improvement over prior work (see Table 2 “missed notes”). We elaborate on this in Section 5 “Limitations”.
>
> **Errors near boundaries:**
> Errors near boundaries may be mislabeled, but this could be mitigated with a sliding window or memory mechanism. We elaborate on this in Section 5 “Limitations”.
>
> **Extreme tempo deviations:**
> The model is designed to be robust to local tempo deviations but is not intended to align performances that diverge dramatically in tempo (like 2x slower), although in these cases a lightweight pre-alignment step can be inserted. This is mainly a practical design choice because in realistic tutoring settings, we expect users to practice near a chosen tempo. We elaborate on this in Section 5 “Limitations”.
>
> ### Questions
>
> > The authors mention “20 beginner performances” but do not specify: a) The number of unique players (only “three graduate students”) vs. performances (20 total — overlap unclear). b) The recording conditions (microphone type, room noise, instrument consistency).
>
> The pieces were evenly distributed among the 3 players (6, 7, and 7 pieces, respectively). Details on unique players are provided in Section 4.1 and Appendix A.8.1.
>
> We apologize for the omission of our recording conditions. The updated manuscript now contains them (Appendix A.8.1). In brief, the dataset was acquired by capturing the direct audio output from a digital piano setup, which utilized sounds from a Yamaha Disklavier.
>
> > How long are the symbolic prompts relative to the output sequence (do they truncate or pad)? Does the model ever ignore the prompt when symbolic and audio information disagree? Was any ablation done on partial or noisy symbolic prompts (realistic for imperfect MIDI scores)?
>
> The symbolic prompts are truncated to represent the same-length segment as the score audio.
>
> Regarding disagreement or noisy scores, in our current task definition, this scenario is actually impossible. As shown in Figure 2, both the reference score audio and the symbolic prompt are generated from the same known, correct reference score. They are two different representations of the same reference. We agree, however, that in alternative settings (e.g., using imperfect symbolic scores from transcription), this would be a failure point. We recommend, in that case, simply sticking with the ladder encoder (audio-only), as it brings about large performance improvements already.

---

### Official Review · Reviewer_gi3B · 2025-11-02

**Soundness:** 4
**Presentation:** 3
**Contribution:** 4
**Rating:** 8
**Confidence:** 3

**Summary:**

The paper introduces LadderSym, a multimodal Transformer architecture for music practice error detection, aiming to identify missed, extra, and wrong notes by comparing student practice audio to a reference score. LadderSym addresses these with two key innovations:

1. A two-stream “Ladder” encoder that incorporates inter-stream alignment modules at every layer, enhancing fine-grained alignment between score and performance audio.

2. A symbolic-score prompting strategy, where symbolic (MIDI) tokens are fed to the decoder, improving interpretability and reducing ambiguity.

**Strengths:**

• Proposes a novel interleaved encoder with layer-wise cross-attention modules, representing a meaningful architectural advancement over late-fusion approaches like Polytune.

• Integrates symbolic prompts—a creative use of decoder conditioning that blends multimodal reasoning with interpretability.

• The model is well-motivated and rigorously validated: strong baselines (Polytune, DTW-based), comprehensive ablations (fusion depth, input modality), and statistically meaningful improvements across datasets.

• The paper is well-organized and accessible, explaining design intuitions (e.g., balancing local vs global feature specialization) with illustrative figures and tables.

**Weaknesses:**

• It would strengthen claims of generality to include instruments with continuous pitch (e.g., strings, voice) or evaluate robustness to stylistic variation.

• The system’s inference latency (8.1 tokens/s) may hinder real-time feedback applications in tutoring contexts. A short section on runtime optimization or lightweight variants would make the work more actionable.

**Questions:**

see weakness

---

> ### Author Response · Authors · 2025-11-20
>
> We thank the reviewer for their detailed review. Below, we address overall concerns and questions:
> ### Weaknesses
> > It would strengthen claims of generality to include instruments with continuous pitch (e.g., strings, voice) or evaluate robustness to stylistic variation.
>
> **On instruments with continuous pitch (e.g., strings, voice):**
> We would like to clarify that our CocoChorales-E dataset *does* include string instruments and their inherent micro-variations (such as vibrato and slides). We provide detail on this in Appendix A.11, Table 9. We observe that at a similar data count, continuous pitch instruments (Violin, Viola, Cello, Contrabass) have similar performance to those of discrete pitch (Oboe, Bassoon, Horn, Tuba). Our paper *does not* study singing voices, and we leave this to future work.
>
> **On robustness to stylistic variation:**
> We agree this is an important point for systems aiming to support intermediate and expert performance. To date, music error detection has primarily aimed at early-stage music learners, where users are often focused on fundamentals like muscle memory and sight-reading rather than developed artistic styles. Our current work advances the state of the art within that scope, and so we do not focus on robustness to highly expressive performance variations.
>
> > The system’s inference latency (8.1 tokens/s) may hinder real-time feedback applications in tutoring contexts. A short section on runtime optimization or lightweight variants would make the work more actionable.
>
> We thank the reviewer for this important point regarding latency in real-time applications. We apologize for the confusion caused by this 8.1 tokens/s metric. We introduced that metric to permit a relative performance comparison to the state-of-the-art rather than an absolute indication of real-time performance. Let us clarify the situation for the reviewer’s consideration.
>
> First, our model is definitely sufficient for real-time operation. The end-to-end latency is under **300ms** to process 2 seconds of audio, and would thus take less than 5 seconds to process a 30-second passage—a long length for a beginner musician.
>
> Furthermore, the token-per-second metric is imprecise, because note density (i.e., number of tokens per frame) varies with each frame of audio. Our model processes music in frames, and the main latency comes from the encoder rather than the autoregressive generation of tokens. Thus, a frame with higher note density would have a higher tokens/s simply because that frame would generate more tokens for a single fixed encoding cost. To avoid this input dependency, our measured tokens/sec in the paper are **worst-case measurements** (if a frame only outputs 1 token). This is meant to only illustrate our relative improvements over Polytune and does not serve as an accurate characterization of our model’s true speed. Even taking this 8.1 tokens/s metric at face value, such a speed would be able to stream passages consisting of 16th notes at 120bpm (as one note onset roughly corresponds to 1 token).
>
> For transparency, we measure the latency on an A100 of each model stage, averaged over 5000 runs, which provides a more accurate characterization of absolute performance:
>
> | Model | Encoder latency (s) | Decoder 1st Token latency (s) | Decoder 100 Tokens latency (s) | Worst Case Token latency (ms) |
> | :--- | :--- | :--- | :--- | :--- |
> | **Polytune** | 0.129 ± 0.024 | 0.00786 ± 0.0356 | 0.7849 ± 0.0057 | 136.86 ± 0.0596 |
> | **LadderSym** | **0.0971 ± 0.0398** | **0.00787 ± 0.0201** | **0.8212 ± 0.0096** | **104.97 ± 0.0599** |
> | **Ladder** | 0.0972 ± 0.0452 | 0.00801 ± 0.0364 | 0.8036 ± 0.0080 | 105.21 ± 0.0816 |
>
> This table shows that LadderSym is capable of generating 100 tokens/s if one frame has such high note density. In realistic use, a 2s frame produces roughly 20 tokens, so decoding takes about 200 ms.
>
> What matters for real-time operation is per-block latency: in real-time DSP, a system is real-time if each block is processed before the next one arrives [1]. On an A100, LadderSym processes a 2s frame in ≈300 ms (≈100 ms encoder + ≈200 ms decoder), which remains well below the 2s block period even after adding a conservative budget of ≈200 ms for network and client overhead [2]. If lower latency were desired, the same model could be run with a sliding window, which would further reduce user-perceived delay and make the application more interactive.
>
> Finally, regarding the suggestion for optimization, our architectural improvements (as indicated in Section 4.3) actually reduce runtime. LadderSym has fewer parameters (172M vs. 192M) and faster inference (8.1 vs. 6.8 tokens/s, almost a 20 percent improvement) due to the improved encoder.
>
> **References:**
> [1] Kuo, S. M., Lee, B. H., and Tian, W. *Real-Time Digital Signal Processing: Implementations and Applications*. Wiley, 2006.
> [2] Kaup, F et.al, D. Assessing the implications of cellular network performance on mobile applications. In *IEEE LCN*, 2016.

---

### Meta-Review · Area_Chair_dUah · 2026-01-07

**Summary:**

The reviewers all see this paper as an interesting and well-executed contribution to music practice error detection. The proposed architecture is assessed to be a meaningful architectural advance over previous work based on DTW or late-fusion approaches. Strengths include the interleaved two-stream encoder, symbolic prompting, careful ablations, and strong empirical validation. Overall sentiment is positive; multiple reviewers describe the work as well-motivated and practically relevant. While not emphasized by the paper, the proposed LadderSym architecture could be of interest to the broader ICLR audience beyond music applications.

**Reviewer Concerns:**

The rebuttal addresses most concerns raised by reviewers. Questions about generalization to continuous-pitch instruments and real-time feasibility are clarified with additional dataset details and latency analysis. Remaining concerns are relatively minor and forward-looking: robustness to highly expressive performances, interpretability of error-specific attention patterns, whether the symbolic prompting contribution is as strong as the ladder architecture itself. None of these concerns are fundamental to the core contributions of this paper.

**Reviewer Scores:**

gi3B: likely unchanged (already positive)

3opm: unchanged (already quite positive)

bM19 – concerns are largely resolved by rebuttal; score could move upward if bM19 is receptive

BK2b – possible upward move; concerns about significance of contributions beyond the Polytune late-fusion may remain

---

### Decision · Program_Chairs · 2026-01-26

Accept (Poster)